# Effectiveness and trend forecasting of tuberculosis diagnosis after the introduction of GeneXpert in a city in south-eastern Brazil

**Thaís Zamboni Berra** [1] * , **Dulce Gomes** [2] , **Antônio Carlos Vieira Ramos** [1‡], **Yan Mathias Alves** [1‡], **Alexandre Tadashi Inomata Bruce** [1‡], **Luiz Henrique Arroyo** [1‡], **Felipe Lima dos Santos** [1‡], **Ludmilla Leideanne Limirio Souza** [1‡], **Juliane de Almeida Crispim** [1‡], **Ricardo Alexandre Arcêncio** [1]

1 Department of Maternal-Infant and Public Health Nursing, University of São Paulo at Ribeirão Preto College of Nursing, Ribeirão Preto, São Paulo, Brazil, 2 Mathematics Department in University of Évora, Évora, Portugal

☯ These authors contributed equally to this work.
‡ These authors also contributed equally to this work.
* thaiszamboni@live.com

**Data Availability Statement:** All relevant data are within the manuscript and its Supporting Information files.

## Abstract

### Background

To evaluate the effectiveness of a rapid molecular test for the detection of tuberculosis (TB) and to predict the rates of disease in a municipality of Brazil where TB is endemic.

### Methods

An ecological study was carried out in Ribeirão Preto-SP on a population of TB cases notified between 2006 and 2017. Monthly TB incidence rates and the average monthly percentage change (AMPC) were calculated. In order to identify changes in the series, the breakpoint technique was performed; the rates were modelled and predictions of the incidence of TB until 2025 were made.

### Results

AMPC showed a fall of 0.69% per month in TB and human immunodeficiency virus (TB-HIV) co-infection, a fall of 0.01% per month in general and lung TB and a fall of 0.33% per month in extrapulmonary TB. With the breakpoint technique, general and pulmonary TB changed in structure in late 2007, and extrapulmonary TB and TB-HIV co-infection changed in structure after 2014, which is considered the cut-off point. The IMA(3) models were adjusted for general and pulmonary TB and TB-HIV co-infection, and the AR(5) models for extrapulmonary TB, and predictions were performed.

### Conclusions

The rapid molecular test for TB is the method currently recommended by the WHO for the diagnosis of the disease and its main advantage is to provide faster, more accurate results and to already check for drug resistance. It is necessary that professionals encourage the

**Funding:** TZB received funding from the São Paulo State Research Support Foundation (FAPESP) [Process FAPESP n˚ 2018 / 03700-7] and RAA received funding from the National Council for Scientific and Technological Development (CNPq) - Scholarship research productivity [Process No. 04483 / 2018-4].

**Competing interests:** The authors have declared that no competing interests exist.

use of this technology in order to optimize the diagnosis so that the treatment begins as quickly as possible and in an effective way. Only by uniting professionals from all areas with health policies aimed at early case identification and rapid treatment initiation it is possible to break the chain of TB transmission so that its rates decrease and the goals proposed by the WHO are achieved.

## Background

Tuberculosis (TB) is a disease that affects millions of people annually. It is classified as one of the most lethal infectious diseases in the world and it is the main cause of death among people diagnosed with HIV (human immunodeficiency virus), with the number of deaths from TB greater than those caused by AIDS (Acquired Immunodeficiency Syndrome) today [1].

In 2019, the World Health Organization (WHO) estimated that there were ten million new cases of TB in the world, with 56% of these cases being in men, 32% in women and 12% in children under 15 years of age; among those, 8.2% were people living with HIV. It is also estimated that there were 1.4 million deaths from TB and 208,000 deaths due to co-infection by TB and human immunodeficiency virus (TB-HIV). In Brazil, approximately 74,000 new cases were diagnosed in 2019, with more than 13,000 cases of relapse of the disease, and about 500 cases of drug addiction [1].

Regarding the diagnostic technologies available in the Brazilian scenario, it is important to highlight that smear microscopy is the routine test in the Brazilian health system (called Sistema Único de Saúde–SUS), and that a culture test is considered as the gold standard for identifying the disease [2]; however, because of the delay in obtaining results due to the prolonged turnaround time of *Mycobacterium tuberculosis* (on average four to eight weeks) and the high complexity, a culture is rarely used for making decisions related to the treatment of a person affected by the disease [3,4]. In addition, it is worth noting that for the diagnosis of TB, imaging examination, clinical evaluation and histology or cytology can also be considered for cases of extrapulmonary TB.

After almost a century of stagnation regarding TB diagnostic techniques, the WHO, in 2010, approved the use of the GeneXpert® MTB/RIF system to perform rapid molecular testing for TB, which is a molecular method based on the real-time PCR (polymerase chain reaction). This is an amplification test for nucleic acids which is used to detect the DNA of *M. tuberculosis* and to screen strains that are resistant to rifampicin, one of the main antibiotics used in the treatment of TB [5].

In a survey of the scientific literature conducted in the main databases, no studies were found that assessed the impact of the rapid molecular test for TB for the detection of cases of disease in the routine activities of health teams. However, studies show that the test performed by the GeneXpert® MTB/RIF system is highly sensitive to pulmonary TB (transmissible form of the disease) [6] and is also the most cost-effective [7–9], as it provides faster results without requiring sample treatment or specialized human resources, thus allowing treatment to begin immediately [10]. However, despite this good cost-benefit, it is worth noting that, in general, it is not a technology considered to be low cost, which can be considered a major disadvantage for developing countries.

For this reason, this study aimed to evaluate the effectiveness of GeneXpert® MTB/RIF for the detection of pulmonary TB, extrapulmonary TB and TB-HIV co-infection and to predict the rates of the disease in the coming years if the routine activities of the health teams in a municipality of south-eastern Brazil are maintained.

## Methods

### Research design and scenario

This was an ecological study [11] carried out in Ribeirão Preto, a municipality in the interior of São Paulo. With regard to the care of people in the municipality with TB, the Basic Health Units (BHSs) are responsible for carrying out active searches for respiratory symptoms, with the collection of sputum smear microscopy and/or X-ray requests. However, the treatment and follow-up of TB cases is performed in specialized outpatient clinics for infectious patients [12].

It is noteworthy that the rapid molecular test for TB in Ribeirão Preto, which involves using the GeneXpert® MTB/RIF system, was implemented and started to be used as a diagnostic technology for TB in November 2014; this was the cut-off point considered in the study.

### Population

The study population consisted of TB cases notified to the Tuberculosis Patient Control System (TBWeb) from 2006 to 2017, which were made available through the Epidemiological Surveillance Division of the Ribeirão Preto Municipal Secretariat.

In the state of São Paulo, a decision has been made to use a single system for the notification and monitoring of people with TB. In TBWeb, which started to be used effectively from 2006, notifications are made online; the main advantage of this system is the uniqueness of each patient's records, and the automatic communication in cases of transfer and hospitalization [13].

Confirmed cases of TB among individuals residing in Ribeirão Preto were considered. Only one record per person was adopted as the selection criterion, with the most current record being selected if there was more than one entry in the system.

The notified TB cases were separated in order to show the differences in the time series for different groups in the municipality: general TB, pulmonary TB, extrapulmonary TB and TB-HIV co-infection.

### Analysis plan

**Calculation of incidence rates.** The monthly incidence rates of TB in the municipality were calculated as the number of cases per month (corrected by the number of days in that month) divided by the corresponding estimated population for the investigation period (2006 to 2017), thus resulting in 144 monthly time observations [14]. Subsequently, the Average Monthly Percentage Change (AMPC) of incidence rates was calculated, identifying, in average percentage terms, any increase or decrease in rates during the study period.

**Detection of structural changes.** In an analysis of time series, in addition to the usual variability observed over time, the data can also be influenced by various types of event that can cause structural changes. Thus, in order to identify possible changes in the series, the R CRAN package *strucchange* was used [15].

Basically, a $y_i$ series is considered and it is assumed that there are $m$ breakpoints in the series, in which the coefficients change from one stable regression relationship to another. There are, therefore, $m + 1$ segments in which the regression coefficients are constant and the model can be rewritten as:

$$y_t = x_t^{\tau}\beta_j + u_t \qquad (t = t_j + 1, \ldots, t_j, \; j = 1, \ldots, m + 1)$$

with $x_t$ being the vector of the co-variables, $\beta_j$ (where $j$ denotes the segment index) the

corresponding regression coefficients, and $u_t$ white noise (that is, an uncorrelated series, with zero mean and constant variance) [16].

**Modelling and forecasting future values.** To model the TB rates, and to predict their future values, incidence rates smoothed by first-order moving averages were considered, as proposed by Becketti [17], using the model for linear time series called the autoregressive integrated moving average (ARIMA) model. The analysis steps proposed by Box and Jenkins [18] were adapted for the chosen model, based on the data structure itself: Identification, Estimation, Verification and Forecasting.

An ARIMA model $(p, d, q)$ allows the variability of a time-related, linear, stationary $(d = D = 0)$ or non-stationary (otherwise) process to be described.

The letters $p$ and $q$ represent, respectively, the number of parameters of the autoregressive parts and the moving averages within the period, and the letter $d$ represents the degrees of simple differentiation necessary to transform a non-stationary series into a stationary one [19].

An ARIMA model can be written as follows:

$$\Delta(\mathrm{B}^s)\Phi(\mathrm{B})(1-\mathrm{B})^d(1-\mathrm{B}^s)^D T(X_t) = \Psi(\mathrm{B}^s)\Theta(\mathrm{B})Z_t$$

where:

$$\Phi(\mathrm{B}) = 1 - \phi_1\mathrm{B} - \phi_2\mathrm{B}^2 - \cdots - \phi_p\mathrm{B}^p, \Theta(\mathrm{B}) = 1 - \theta_1\mathrm{B} - \theta_2\mathrm{B}^2 - \cdots - \theta_q\mathrm{B}^q$$

are, respectively, the autoregressive and the moving average polynomials. $T$ is the transformation to stabilize the variance, if this is necessary, and $Z_t$ represents the white noise process (a non-correlated process, with zero mean and constant variance).

The KPSS unit root test was performed to determine whether the series was stationary or not, using a significance level of 5%. For a non-stationary series, it is necessary to resort to the usual transformation techniques (Box–Cox and simple differentiations) in order to transform the series into a stationary one and then to determine, through the empirical autocorrelation and partial autocorrelation functions, the p and orders of the ARIMA model.

To estimate the model parameters, the maximum likelihood method was used. For the validation of the model, namely in the analysis of residues, the usual tests of absence of autocorrelation (Portmanteau tests: Ljung–Box and Box–Pierce), randomness (Rank and Turning Point tests), normality (Kolmogorov–Smirnov test) and the t test of average nullity were performed.

It is worth mentioning that the choice of the best model was made taking into account the lowest values of the Akaike information criterion (AIC). Subsequently, data and trend forecasts were made for an eight-year period (2018 to 2025).

A set of tests using the last two years (2016 to 2017) was used to assess the predictive performance of the models. The following measures were considered in order to evaluate this predictive performance: Root Mean Square Error (RMSE), which indicates the difference between the values predicted by a model and the observed values, Mean Absolute Error (MAE), which is a measure of the accuracy of a forecast in the estimation of trends, and Mean Absolute Percentage Error (MAPE), which is the percentage of the predicted values that are incorrect. Forecasts were then made based on the adjusted models for the eight-year period (2018 to 2025).

We have chosen to present possible trajectories for the forecasts, instead of the usual average forecasts (since when the series are stationary or show trends and/or seasonality, the forecasts do not usually tend towards the process average), through a simulation of the adjusted model. Both for the simulation of future trajectories and for the calculation of the respective confidence intervals of the predictions, errors were considered as random variables, normally distributed, with a mean and standard deviation equal to their estimated values.

All of the analyses were performed using the statistical software R Studio® version 3.5.2 (https://rstudio.com).

### Ethics approval and consent to participate

In compliance with Resolution 499/2012 of the National Health Council, the study was approved by the Research Ethics Committee of Nursing College of Ribeirão Preto, University of São Paulo under the Certificate of Ethical Appraisal number 87696318.3.0000.5393 issued on April 30, 2019. Consent to participate was not applicable, because the work was performed using secondary data from cases diagnosed with TB and reported on TBWeb.

## Results

Between 2006 and 2017, 2259 TB cases were reported in Ribeirão Preto, the majority of which were pulmonary (77.9%). Fig 1 shows the time series of incidence rates per 100,000 inhabitants.

Table 1 shows the frequency of distribution of TB cases grouped by type, the annual rate and the AMPC.

Fig 1 shows a decrease in the incidence rate of TB-HIV co-infection, which can be proved through the AMPC, showing a decrease of 0.69% per month, as seen in Table 1. It is also possible to verify that general and pulmonary TB decreased by 0.01% per month, and that extrapulmonary TB decreased by 0.33% per month, although these reductions are less noticeable when analyzed in Fig 1.

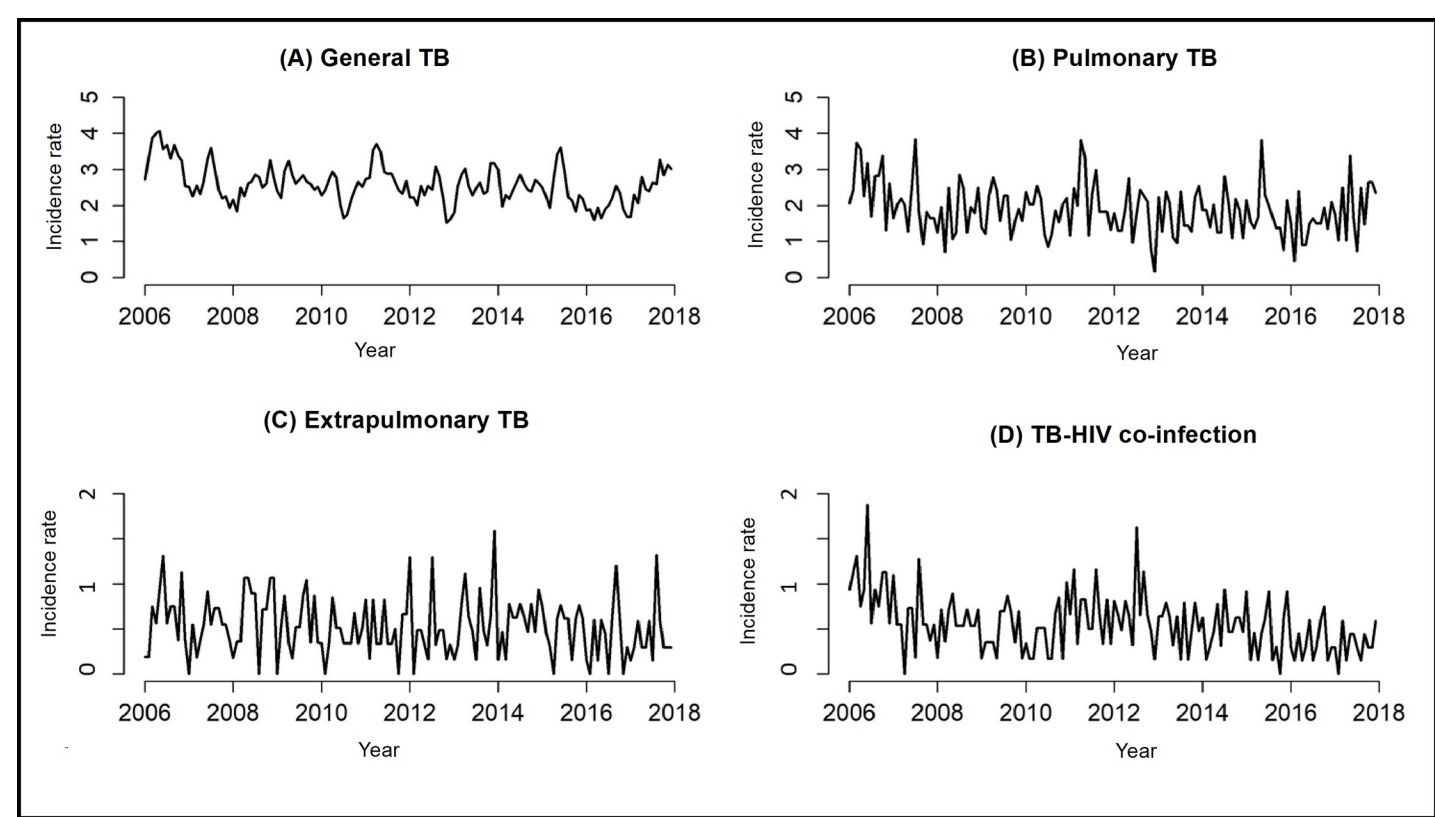

**Fig 1. Tuberculosis incidence rates in Ribeirão Preto, São Paulo, Brazil (2006–2017).**

**Table 1. Profile of tuberculosis cases according to classification and average monthly percentage variation of rates, Ribeirão Preto, São Paulo, Brazil (2006–2017).**

| Tuberculosis | Absolute frequency (%) | Annual rate (100,000 inhab.) | Average Monthly Percentage Change (%) |
|---|---|---|---|
| General | 2259 (100%) | 31.7 | -0.01 |
| Pulmonary | 1760 (77.9%) | 23.2 | -0.01 |
| Extrapulmonary | 497 (22.0%) | 6.4 | -0.33 |
| TB-HIV co-infection | 510 (22.6%) | 6.7 | -0.69 |

With the breakpoint technique, it is possible to identify the time series structure changes. For general and pulmonary TB, the series structure changed in late 2007, whereas the series for the incidence rates of extrapulmonary TB and TB-HIV co-infection show a break after 2014, which is the cut-off point considered in this study because it marks the beginning of TB diagnosis using the rapid molecular test for TB. Fig 2 shows the breakpoint technique and the various confidence intervals.

With the KPSS unit root test, a value of 0.65 was obtained for general TB (the series is non-stationary), 0.65 for pulmonary TB (non-stationary series), 0.37 for extrapulmonary TB (stationary series) and 1.03 for TB-HIV co-infection (non-stationary series). Thus, it was necessary to use the transformations for the series identified as non-stationary.

For general TB, pulmonary TB and TB-HIV co-infection, the best adjusted models were of ARIMA type (0,1,3), or simply IMA(3). For general TB, only the constant and the coefficient

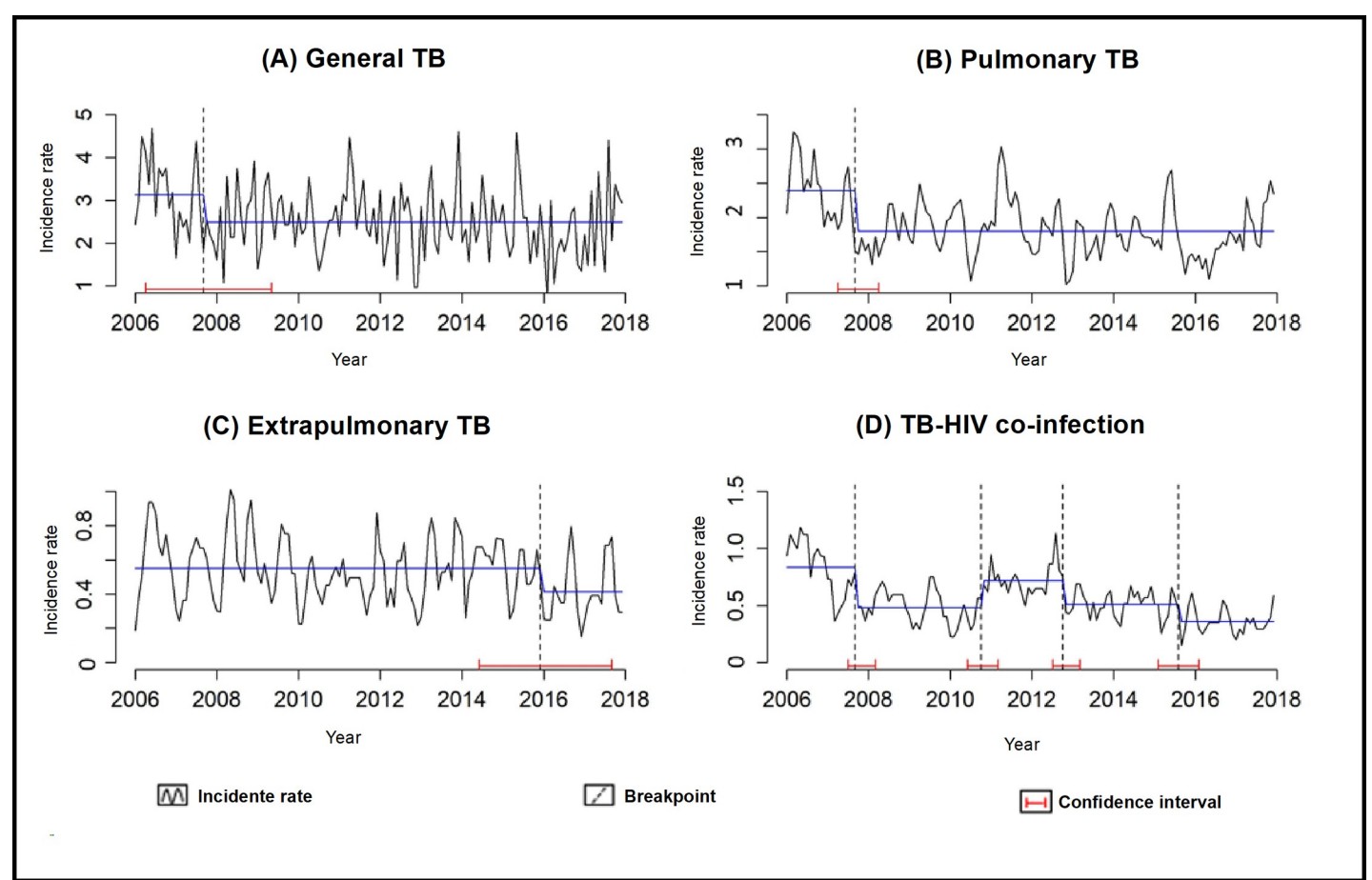

**Fig 2. Changes in the structure of the time series regarding the incidence rates of tuberculosis in Ribeirão Preto, São Paulo, Brazil (2006–2017).**

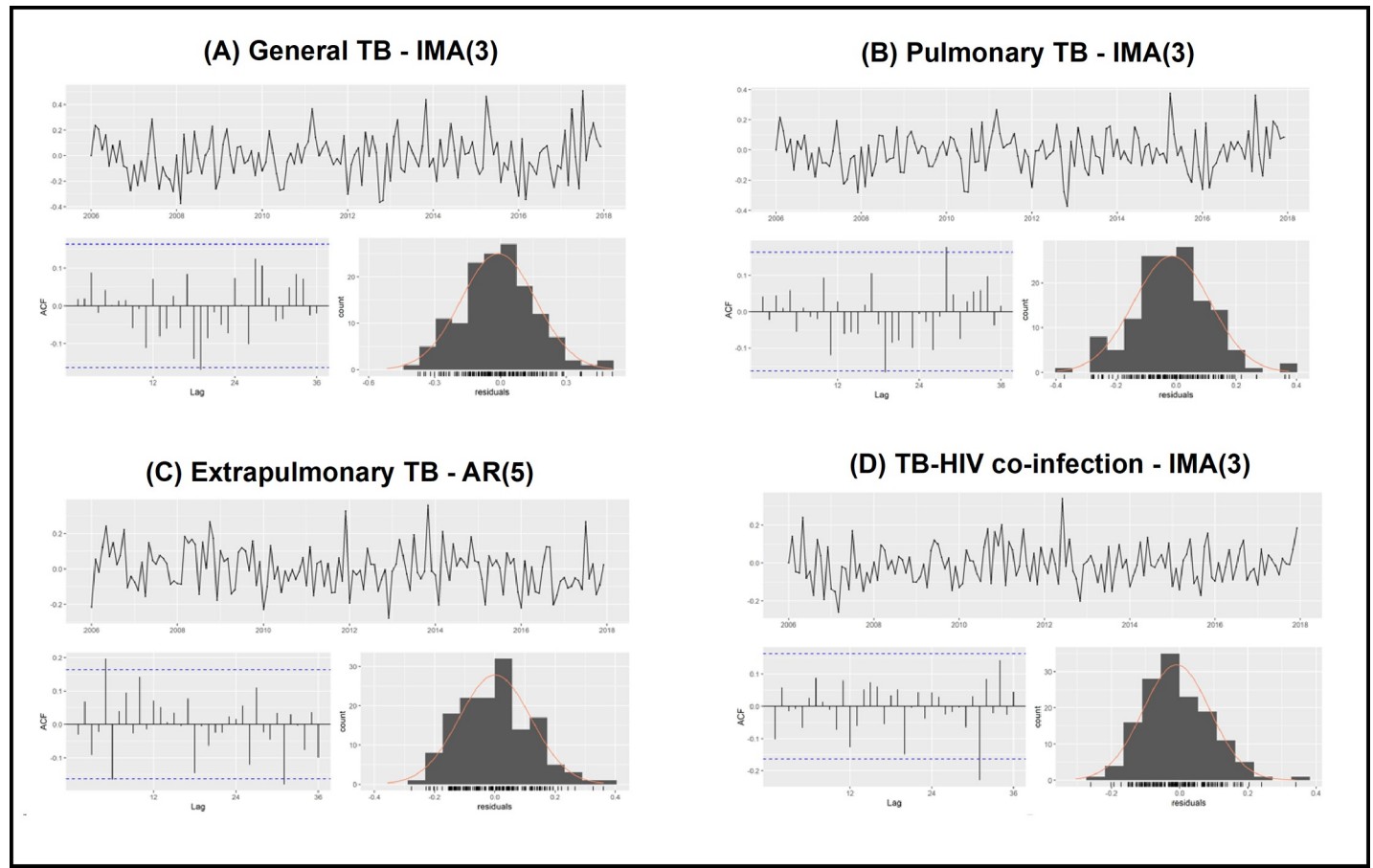

**Fig 3. Residue analysis (schedule, ACF and histogram) of the models estimated for the tuberculosis rates, Ribeirão Preto, São Paulo, Brazil, 2006–2017.**

of order three were significant; for pulmonary TB, only the coefficient of order three was significant; and for TB-HIV co-infection, the coefficients of orders two and three were significant. For extrapulmonary TB, the best model found was an ARIMA(5,0,0), or simply <u>the</u> AR(5) model, with all the coefficients being significant, except for the coefficient of order two.

After choosing the best models, we proceeded to analyze the residues. It is possible to observe (Fig 3 and Table 2) that the residues are statistically unrelated and that the remaining

**Table 2. Analysis of residues from the temporal modeling of tuberculosis rates, Ribeirão Preto, São Paulo, Brazil (2006–2017).**

| | Test Statistics (p-value) | | | |
|---|---|---|---|---|
| **Test** | **General TB** | **Pulmonary TB** | **Extrapulmonary TB** | **TB-HIV co-infection** |
| Ljung-Box | 0.04 (0.83) | 0.24 (0.61) | 0.14 (0.70) | 1.54 (0.21) |
| Box-Pierce | 0.044 (0.83) | 0.24 (0.62) | 0.13 (0.70) | 1.51 (0.21) |
| Rank test | 0.82 (0.40) | 0.82 (0.40) | -2.16 (0.30) | 0.97 (0.33) |
| Turning Point | -0.33 (0.74) | -0.92 (.035) | 1.06 (0.28) | 0.66 (0.50) |
| Difference Sign Test | 0.14 (0.88) | -0.71 (0.47) | -1.00 (0.31) | -0.71 (0.47) |
| Bartlett B test | 0.45 (0.98) | 0.57 (0.90) | 0.95 (0.31) | 0.76 (0.59) |
| Kolmogorov–Smirnov | 0.04 (0.92) | 0.04 (0.96) | 0.04 (0.88) | 0.06 (0.60) |
| T test of means | -0.63 (0.52) | -1.32 (0.18) | 0.07 (0.94) | -1.10 (0.27) |

**Table 3. Predictive analysis of the TB incidence according the clinical condition of the cases, Ribeirão Preto, São Paulo, Brazil (2006–2017).**

| TB | RMSE | MAE | MAPE |
|---|---|---|---|
| General TB | 0.27 | 0.21 | 8.5 |
| Pulmonary TB | 0.23 | 0.18 | 9.81 |
| Extrapulmonary TB | 0.11 | 0.09 | 19.41 |
| TB-HIV co-infection | 0.09 | 0.07 | 15.31 |

assumptions of the models are validated (independent and identically distributed residues, with a normal distribution of zero mean and constant variance), considering a significance level of 5%.

The quality of the forecasts was analyzed by comparison with the subset of tests (years 2016 and 2017) as seen in Table 3.

Fig 4 shows the final models for the series; it is possible to observe that all of the adjustments are able to capture the variability of all of the series under analysis in a very satisfactory way.

## Discussion

The study aimed to assess the effectiveness of GeneXpert$^®$ MTB/RIF for the detection of pulmonary TB, extrapulmonary TB and TB-HIV co-infection and to predict the rates of disease in the coming years if the routine activities of health teams in this municipality of south-eastern Brazil are maintained. Among the limitations of the study includes the use of secondary data sources, which can lead to incomplete data or typos.

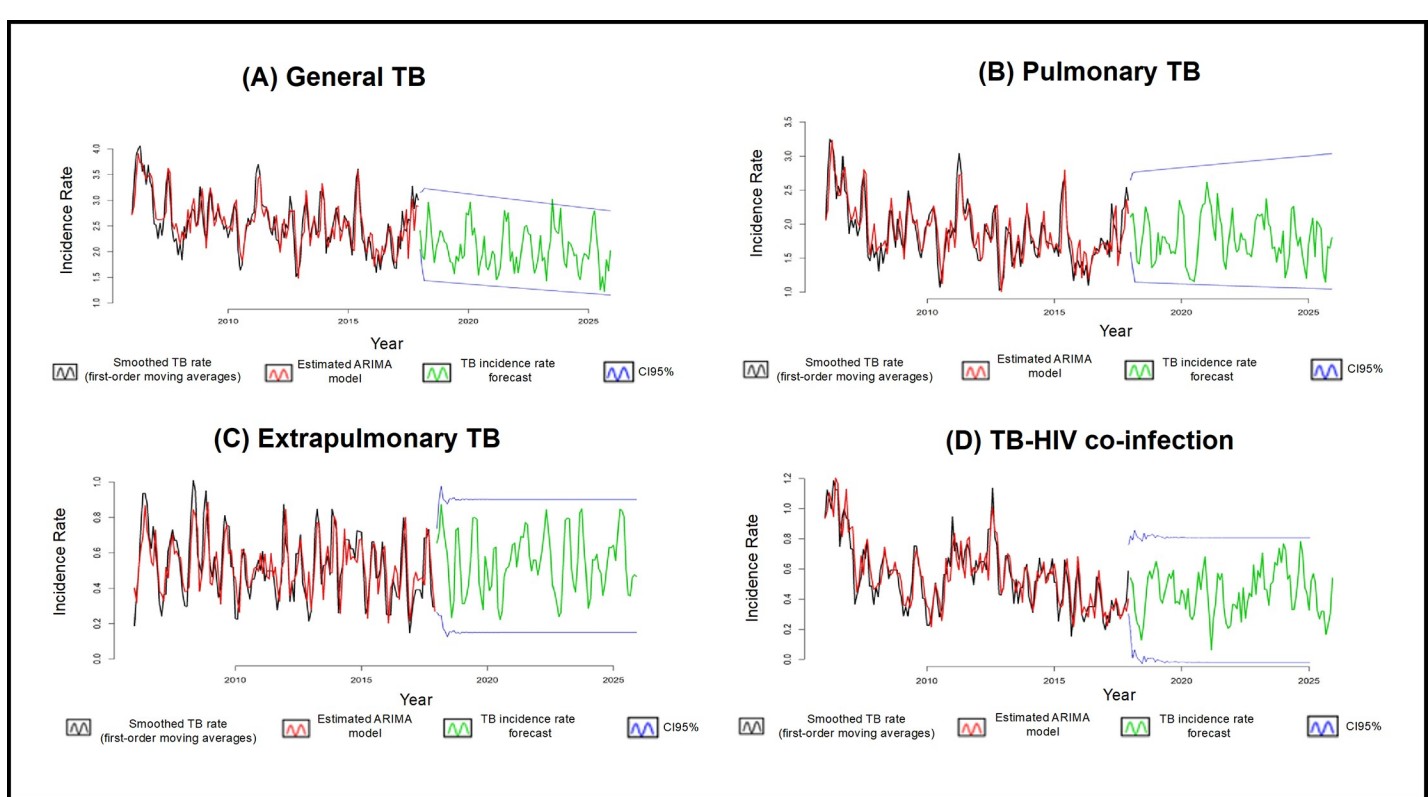

**Fig 4. Models adjusted for tuberculosis rates (2006–2017), forecast and respective 95% confidence intervals (2018–2025), Ribeirão Preto, São Paulo, Brazil.**

With the analysis of AMPC (Table 1), it is possible to note that, among the groups analyzed in the present study, TB-HIV co-infection showed the greatest decrease (0.69% per month). This can be explained by the greater sensitivity of the rapid molecular test for TB.

Studies show that for a sputum sample in which sputum smear and culture showed positive results, the sensitivity of the rapid molecular test for TB can vary between 98 and 100%. For cases in which the sputum sample has a negative result and a positive culture, the sensitivity of the rapid molecular test for TB decreases to 57 to 78%. For people with negative clinical examination and who had sputum samples with a positive culture result, the specificity of the rapid molecular test for TB can vary between 91% and 100% [2].

For cases in which the culture result is negative, but the person with suspected TB has a positive clinical examination, the specificity of the rapid molecular test for TB is also about 91% but goes to 100% when the sputum sample it has negative sputum smear and culture and is between 94 and 100% when the sputum sample shows only negative culture [2].

Thus, in view of the findings of the study, in addition to the greater sensitivity of the rapid molecular test for TB, the hypothesis is that this decrease in TB-HIV co-infection rate may be related to the greater adherence of people living with HIV to the antiretroviral therapy or positive changes in drug regimens to decrease viral load; this would result in an increase in the immunity of people living with HIV and, therefore, in preventing *M. tuberculosis* infection [20–22].

With the breakpoint analysis, it was possible to identify that the incidence rates for extrapulmonary TB and TB-HIV co-infection showed a change in the structure of its time series after 2014, which is the cut-off point considered in this study as it marks the beginning of TB diagnosis through the rapid molecular test for TB. This change in the time series may indicate that, among other factors, the diagnosis using the rapid molecular test for TB might have been responsible for causing this change in TB rates in the municipality due to its greater sensitivity and specificity.

Regarding extrapulmonary TB, a study showed that the positivity rates for histology or cytology, GeneXpert® MTB/RIF and culture were 12.88%, 20.59% and 15.82%, respectively. The sensitivity and specificity of the rapid molecular test for TB were almost the same in the pulmonary and extrapulmonary samples (78.2% and 90.4%) and (79.3% and 90.3%), respectively [23], however it is worth noting that when the culture was used as the gold standard, the sensitivity and specificity of GeneXpert® MTB/RIF were determined for pulmonary and extrapulmonary TB.

According to the literature [5], most of the articles which analyzed the effectiveness of the rapid molecular test for TB used sputum samples; the studies that evaluated extrapulmonary samples showed good test performance, but it is worth noting that the test was more sensitive for sputum samples in most studies.

According to a literature review [5], most of the studies that analyzed the effectiveness of the GeneXpert® MTB/RIF assay used sputum samples and although the studies that evaluated extrapulmonary samples have shown satisfactory results [24–28], but it is important that further studies are carried out with this type of material, as this result may vary according to the number of samples tested, the nature of the different extrapulmonary TB samples and also the specific differences that can be found in the different epidemiological contexts, but despite these biases, it is noteworthy that in most studies carried out the GeneXpert® MTB/RIF system pointed out that it was the most sensitive test for pulmonary TB, the transmissible form of the disease; however, with the current data found in the literature, there is now support for the use of this method in extrapulmonary samples.

Therefore, in view of the above, health professionals have a good scientific basis for recommendig the performance of the rapid molecular test for TB in the GeneXpert® MTB/RIF

system for suspected cases of extrapulmonary TB, although other diagnostic tests must also be used to confirm diagnosis accurately.

Another study showed that, in samples of urine and bone or joint fluid and tissue, the rapid molecular test was sensitive (more than 80%), meaning that it has a low chance of resulting in a false positive. In cerebrospinal fluid, pleural fluid and peritoneal fluid, the rapid molecular test was highly specific (98% or more), with a low chance of resulting in a false negative [29]; however, it is worth noting that studies have not shown good efficacy with the GeneXpert® assay in samples of pleural fluid [5,30–32].

Therefore, although there are other techniques for detecting extrapulmonary TB, such as histopathology, the use of the rapid molecular test performed using the GeneXpert® MTB/RIF system is safe and has high sensitivity and specificity when compared to the usual tests.

Across the world, the most commonly used tests for the diagnosis of TB are the sputum smear and sputum culture tests (only for pulmonary TB, in cases of extrapulmonary TB, diagnostic tests are performed with other materials). As it is simpler, faster and less costly, smear microscopy ends up being more commonly used as a routine test by health systems, but it is worth mentioning that it has low sensitivity, especially in cases with low bacillary load [33,34].

The sensitivity of the culture varied between 70% and 90% when the sample of the material was correctly collected, stored and all procedures for carrying out the test were correctly followed by the responsible professional, using a work-intensive method that requires qualified professionals for the examination to be carried out accurately and reliably. It takes about 14 to 60 days to get a result with this technique and due to the long period required to reach a diagnosis, culture is rarely used to make decisions regarding the treatment of a person with TB [33,34]. It is worth mentioning that the most used culture media are solids, but liquid culture performed in an automated system has advantages.

According to the Brazilian report of the National Commission for the Incorporation of Technologies in SUS (CONITEC), for the automated liquid culture, the average proportion of mycobacterial detection obtained was 82% (95% CI, 71% to 90%), and 65% (95% CI 51% to 77%) for culture in solid medium. The average time for detection by the automated liquid culture system was 14 days and for solid culture it was 28 days. Regarding the sensitivity test performed in the automated liquid culture system, in general, a high level of agreement with the method of proportions in solid culture (>90%) was identified. Thus, the use of automated systems could increase the coverage of testing across the country, constituting an important strategy for the diagnosis and control of TB [35].

Still regarding liquid culture, a review study found that the highest sensitivities and negative predictive values were shown by the MGIT-960 System when used alone or in combination with the Lowenstein-Jensen method, with the combined media showing better sensitivity as well as obtaining satisfactory results in the detection of non-tuberculous mycobacteria. The greater specificity and positive predictive values were shown by the Lowenstein-Jensen method alone, which did not produce false positives [36].

In another study carried out in a laboratory in South Africa [37], results point to the high cost of MGIT-960 and its cost-benefit must be carefully evaluated taking into account its sensitivity and high probability of cross contamination, as was also reported in another study [38] that compared solid and liquid media for culture.

It is possible to conclude that the MGIT-960 has a better performance regarding sensitivity, while the Lowenstein-Jensen method has greater specificity; the MGIT-960 system can provide a better isolation rate of mycobacteria from sputum samples from people suspected of having pulmonary TB when compared to culture in a solid medium (Lowenstein-Jensen) [39–41]. Therefore, in accordance with current international guidelines, a combination of solid and

liquid media is recommended, so that the number of positive findings can be significantly increased.

Returning to the rapid molecular test for TB performed using the GeneXpert$^{®}$ system, its main advantages are that the result can be released within two hours and will already indicate whether the patient is resistant or sensitive to rifampicin [42]. Thus, it is possible that the incorporation of the GeneXpert® system into the Brazilian health system may be related to changes in disease rates since more people may be diagnosed due to the high sensitivity, specificity and rapid time of diagnosis of the GeneXpert® system when compared to the usual tests and, therefore, it could also be responsible for a consequent change in the structure of the analyzed time series, since any change in the indexes (be it rate or number of cases) alters its history series over time.

Using the Box–Jenkins [18] methodology, a well-estimated model that provides adequate forecasts, can be a very useful tool to assist with decision-making by both public managers and society [43]. From the adjustment of the models, it was possible to verify and make forecasts for the TB rates in Ribeirão Preto; although the estimates indicate a slight decrease, TB will continue to be a serious public health problem.

Considered a priority by the Brazilian government since 2003, TB has been the subject of several national agreements and has been made a priority of the National Tuberculosis Control Program (NTCP). There has been an increase in the detection of cases through the strengthening of the primary health care services system [44,45].

According to the TB diagnosis algorithm recommended by the Ministry of Health, all new suspected cases of pulmonary TB must be tested using a rapid molecular test. If both MTB and resistance to rifampicin are detected, another confirmatory rapid molecular test should be performed, combined with a culture test and drug susceptibility test, and the patient should be referred for specialist attention. If rifampicin-sensitive MTB is detected, treatment with the basic TB regimen is started, but in addition, culture and sensitivity testing must be performed to check resistance to other drugs used in the treatment [4].

The rapid molecular test for TB performed by the GeneXpert$^{®}$ MTB/RIF system has several positive points when compared to classic methods for TB diagnosis; however, further studies will improve the understanding of this test and contribute to the diagnosis of TB.

In the scenario under investigation, the tests are being carried out with the new systems called GeneXpert$^{®}$ Ultra and GeneXpert$^{®}$ Omni, which are evolutions of the GeneXpert$^{®}$ system used in Brazil and which have been shown to be more sensitive and specific; these may have relevant implications for the health services system and an impact on disease rates, as identified in the present study. However, it is important to understand the changes or impacts that occurred in the epidemiological indicators and health services with the diagnosis of TB being made through the rapid molecular test and its progression [46–48].

The study provides advances in knowledge as it presents the impact of rapid molecular testing for TB on routine health services, and it is important to note that studies of this design are able to support decisions about the maintenance and/or sustainability of technologies in a real, uncontrolled context.

## Conclusions

The rapid molecular test for TB performed by the GeneXpert$^{®}$ MTB/RIF system is the method currently recommended by the Ministry of Health for the diagnosis of TB. Its main advantage is that it gives a faster and more accurate diagnosis when compared to other classic diagnostic methods, such as smear microscopy and culture tests.

It is necessary for managers responsible for the Tuberculosis Control Programs to encourage professionals who deal with people suspected or diagnosed with TB to comply with the

algorithm recommended by the WHO in order to optimize the diagnosis so that the technologies are used correctly, meaning that a more accurate diagnosis is carried out so that the treatment can start as quickly as possible and correctly, taking into account whether the identified strain is resistant to the drugs used or not. Only by uniting professionals from all areas with health policies aimed at early case identification and rapid treatment initiation is it possible to break the chain of TB transmission so that its rates decrease and the goals proposed by the WHO are achieved.

## Supporting information

**S1 Dataset.**
(CSV)

## Acknowledgments

Municipal Health Secretariat of Ribeirão Preto.
　　Municipal Tuberculosis Control Program of Ribeirão Preto.

## Author Contributions

**Conceptualization:** Thaís Zamboni Berra, Dulce Gomes, Ricardo Alexandre Arcêncio.

**Data curation:** Thaís Zamboni Berra.

**Formal analysis:** Thaís Zamboni Berra, Dulce Gomes.

**Investigation:** Thaís Zamboni Berra, Dulce Gomes, Ricardo Alexandre Arcêncio.

**Methodology:** Thaís Zamboni Berra, Dulce Gomes.

**Supervision:** Ricardo Alexandre Arcêncio.

**Writing – original draft:** Thaís Zamboni Berra, Dulce Gomes, Ricardo Alexandre Arcêncio.

**Writing – review & editing:** Thaís Zamboni Berra, Dulce Gomes, Antônio Carlos Vieira Ramos, Yan Mathias Alves, Alexandre Tadashi Inomata Bruce, Luiz Henrique Arroyo, Felipe Lima dos Santos, Ludmilla Leideanne Limirio Souza, Juliane de Almeida Crispim, Ricardo Alexandre Arcêncio.

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
