## [Decision Letter · Decision Letter 0]

8 Jan 2021

PONE-D-20-34732

Effectiveness and trend forecasting of tuberculosis diagnosis after the introduction of GeneXpert® in a city in south-eastern Brazil

PLOS ONE

Dear Dr. Berra,

Thank you for submitting your manuscript to PLOS ONE. After careful consideration, we feel that it has merit but does not fully meet PLOS ONE’s publication criteria as it currently stands. Therefore, we invite you to submit a revised version of the manuscript that addresses the points raised during the review process.

Please submit your revised manuscript. If you will need more time than this to complete your revisions, please reply to this message or contact the journal office at plosone@plos.org. Please include the following items when submitting your revised manuscript:

We look forward to receiving your revised manuscript.

Kind regards,

Frederick Quinn

Academic Editor

PLOS ONE

Journal Requirements:

3. Please upload a copy of Supporting Information Figures  S1-S4  and Tables S1-S3 which you refer to in your text on page 20-21.

Reviewers' comments:

Reviewer's Responses to Questions

**Comments to the Author**

1. Is the manuscript technically sound, and do the data support the conclusions?

Reviewer #1: Partly

Reviewer #2: Yes

2. Has the statistical analysis been performed appropriately and rigorously? 

Reviewer #1: I Don't Know

Reviewer #2: Yes

3. Have the authors made all data underlying the findings in their manuscript fully available?

Reviewer #1: No

Reviewer #2: Yes

4. Is the manuscript presented in an intelligible fashion and written in standard English?

Reviewer #1: No

Reviewer #2: Yes

5. Review Comments to the Author

Reviewer #1: Comments:

1. English needs to be improved throughout the manuscript. There are few instances, where the meaning is not clear. Ist line of Introduction, ‘is the main cause of death among people living with HIV (human immunodeficiency virus) (PLHIV), overcoming AIDS (Acquired Immunodeficiency Syndrome) as the most lethal infectious disease of today[1]’ needs to be changed.

2. Please change the ‘behavior of disease’ to better phrase..

3. We already have WHO TB report of 2020, please quote that instead of WHO (2019).

4.What do you understand by bacilloscopy? Do you mean smear microscopy? Smear microscopy is the most commonly used term. Pl use that.

5. Introduction, line 60, ‘slow reproduction of the bacillus’ is wrong. It is the turnaround time of M. tuberculosis, which is 4-8 weeks, though the generation time is ~ 18-20 hrs. Pl correct this.

6. Introduction, line 66, there is no need to define DNA. It is such a common term.

7. To the best I know, TRM-TB is not very common term in many parts of the world. I will avoid using this term. Please write this in full form.

8. Tuberculosis (TB) shall be abbreviated at the first time and then it shall be written as ‘TB’.

9. In Discussion, lines 257-259, ‘Studies show that for samples with negative smear microscopy, the sensitivity of the TRM-TB for a sputum sample is 72.5% and for three samples it reaches almost 91%[2]’

Please rewrite this sentence.

10. What is SUS? In line 268 same page. I believe,it is city. Then write accordingly.

11. Across the world, sputum smear microscopy and culture are commonly used methods for the detection of pulmonary TB. For the detection of extrapulmonary TB, these methods are not good. In fact, TB diagnosis also relies on other diagnostic modalities, like hisopathology, imaging, IGRAs including molecular tests, like LAMP test, PCR, multiplex PCR, real-time PCR, etc.

12. In Discussion, lines 274, ‘A culture test can detect between 70% and 90% of cases; however, it requires at least 14 to 60 days to obtain a diagnosis’ is not written accurate. In fact, the sensitivity of culture in extrapulonary paucibacillary TB specimens varies from 0-80%.

13. In Discussion, line 297, what do you mean by sensitivity test (ST)? Is it drug susceptibility test (DST)?

14. You need to compare Xpert assay at the Lab level with other routine tests for TB diagnosis like imaging, histopathology, cytology, PCR, real-time PCR, etc.Pl mention these tests in the Introduction as well as Discussion.

Reviewer #2: The manuscript titled as "Effectiveness and trend forecasting of tuberculosis diagnosis after the introduction of GeneXpert in a city in south-estern Brazil" is an interesting topic, and supplied the diagnosis for the care of a person with tuerculosis. The manuscript has reached the publication level of PLoS One, I recommend to accept it. I have no comments to the authors.

6. PLOS authors have the option to publish the peer review history of their article (what does this mean?). If published, this will include your full peer review and any attached files.

Reviewer #1: No

Reviewer #2: No

---

## [Author Response · Author response to Decision Letter 0]

12 Jan 2021

Dear Editor, 

Many thanks for your reply and your reviewers' comments about our manuscript “PONE-D-20-34732 Effectiveness and trend forecasting of tuberculosis diagnosis after the introduction of GeneXpert® in a city in south-eastern Brazil”.

The comments were appropriate to qualify/ improve the manuscript. We have forwarded a letter with the changes made in the manuscript, presenting our response for each comment from reviewers, as requested. Many thanks for the comments.

In the current version presented to the journal, we modified the question regarding data availability. Initially, they were collected at the request and approval of an ethics committee, but for the analyzes performed and presented in the present manuscript, we did not use any variable that could identify the cases included in the study. In this way, as the data is unrestricted, we upload the minimum set of anonymous data to replicate the results of the study, as requested.

Regarding the figures and tables of supporting information, there was a mistake on our part and we apologize, this has already been corrected. The figures and tables that we had placed in the supporting information section are actually the figures and tables present in the manuscript itself. There is only one supplementary document that is the minimum dataset, we have already corrected this error in the manuscript and in the system.

Below, we respond point by point to the comments of Reviewer #1 comments and would like to thank Reviewer #2 for spending time reading and reviewing our manuscript.

Kind regards,

Berra et al.

Reviewers' comments:

Reviewer #1: 

1. English needs to be improved throughout the manuscript. There are few instances, where the meaning is not clear. Ist line of Introduction, ‘is the main cause of death among people living with HIV (human immunodeficiency virus) (PLHIV), overcoming AIDS (Acquired Immunodeficiency Syndrome) as the most lethal infectious disease of today[1]’ needs to be changed.

Thank you for your comment. English was extensively revised throughout the manuscript and the first paragraph of the manuscript was reformulated to facilitate understanding.

2. Please change the ‘behavior of disease’ to better frase.

Thank you for your comment. This term has been changed throughout the manuscript.

3. We already have WHO TB report of 2020, please quote that instead of WHO (2019).

Thank you for your comment. The reference and information have been updated.

4. What do you understand by bacilloscopy? Do you mean smear microscopy? Smear microscopy is the most commonly used term. Pl use that.

Thank you for your comment. Bacilloscopy is the same as smear microscopy. The term "bacilloscopy" was changed throughout the manuscript and replaced with smear microscopy, as suggested.

5. Introduction, line 60, ‘slow reproduction of the bacillus’ is wrong. It is the turnaround time of M. tuberculosis, which is 4-8 weeks, though the generation time is ~ 18-20 hrs. Pl correct this.

Thank you for your comment. This has been fixed.

6. Introduction, line 66, there is no need to define DNA. It is such a common term.

Thank you for your comment. The definition of DNA has been removed as suggested.

7. To the best I know, TRM-TB is not very common term in many parts of the world. I will avoid using this term. Please write this in full form.

Thank you for your comment. Throughout the manuscript, the acronym TRM-TB was changed to rapid molecular test for TB.

8. Tuberculosis (TB) shall be abbreviated at the first time and then it shall be written as ‘TB’.

Thank you for your comment. This has been fixed.

9. In Discussion, lines 257-259, ‘Studies show that for samples with negative smear microscopy, the sensitivity of the TRM-TB for a sputum sample is 72.5% and for three samples it reaches almost 91%[2]’

Please rewrite this sentence.

Thank you for your comment. This sentence has been removed from the discussion and a paragraph has been inserted that further details this information to facilitate understanding.

10. What is SUS? In line 268 same page. I believe,it is city. Then write accordingly.

Thank you for your comment. The acronym SUS stands for Sistema Único de Saúde and refers to the brazilian health system, which had been explained briefly in the introduction. Understanding that it can cause confusion among readers, the acronym SUS was replaced in the entire manuscript by the brazilian health system.

11. Across the world, sputum smear microscopy and culture are commonly used methods for the detection of pulmonary TB. For the detection of extrapulmonary TB, these methods are not good. In fact, TB diagnosis also relies on other diagnostic modalities, like hisopathology, imaging, IGRAs including molecular tests, like LAMP test, PCR, multiplex PCR, real-time PCR, etc.

Thank you for your comment. Three paragraphs were inserted in the discussion of studies proving the effectiveness of the rapid molecular test for tuberculosis in extrapulmonary samples. Although there are other diagnostic techniques as you mentioned, GeneXpert® MTB / RIF is a reliable option for this type of sample.

12. In Discussion, lines 274, ‘A culture test can detect between 70% and 90% of cases; however, it requires at least 14 to 60 days to obtain a diagnosis’ is not written accurate. In fact, the sensitivity of culture in extrapulonary paucibacillary TB specimens varies from 0-80%.

Thank you for your comment. This information regarding the sensitivity of the culture was based on the references cited in this paragraph. In addition, I did a search and in addition to those mentioned, other articles and government documents (from Brazil and the world) reinforce this information that the sensitivity of the culture can vary between 70 and 90% when the sample was correctly collected, stored and the technique the exam was correctly performed. I think this may have caused some confusion. The sentence has been reformulated to improve understanding.

13. In Discussion, line 297, what do you mean by sensitivity test (ST)? Is it drug susceptibility test (DST)?

Thank you for your comment. Yes, the sensitivity test is the drug susceptibility test. The term was changed in the manuscript to facilitate understanding.

14. You need to compare Xpert assay at the Lab level with other routine tests for TB diagnosis like imaging, histopathology, cytology, PCR, real-time PCR, etc.Pl mention these tests in the Introduction as well as Discussion.

Thank you for your comment, this information was included in the introduction and discussion

Reviewer #2: 

The manuscript titled as "Effectiveness and trend forecasting of tuberculosis diagnosis after the introduction of GeneXpert in a city in south-estern Brazil" is an interesting topic, and supplied the diagnosis for the care of a person with tuerculosis. The manuscript has reached the publication level of PLoS One, I recommend to accept it. I have no comments to the authors.

I would like to thank you for having read and evaluated the manuscript and it was very gratifying for us to know that, in your opinion, it is at the level of publication in such a prestigious journal.

---

## [Decision Letter · Decision Letter 1]

22 Feb 2021

PONE-D-20-34732R1

Effectiveness and trend forecasting of tuberculosis diagnosis after the introduction of GeneXpert® in a city in south-eastern Brazil

PLOS ONE

Dear Dr. Berra,

Thank you for submitting your manuscript to PLOS ONE. After careful consideration, we feel that it has merit but does not fully meet PLOS ONE’s publication criteria as it currently stands. Therefore, we invite you to submit a revised version of the manuscript that addresses the points raised during the review process.

Please submit your revised manuscript. If you will need significantly more time to complete your revisions, please reply to this message or contact the journal office at plosone@plos.org. Please include the following items when submitting your revised manuscript:

We look forward to receiving your revised manuscript.

Kind regards,

Frederick Quinn

Academic Editor

PLOS ONE

Reviewers' comments:

Reviewer's Responses to Questions

**Comments to the Author**

1. If the authors have adequately addressed your comments raised in a previous round of review and you feel that this manuscript is now acceptable for publication, you may indicate that here to bypass the “Comments to the Author” section, enter your conflict of interest statement in the “Confidential to Editor” section, and submit your "Accept" recommendation.

Reviewer #1: (No Response)

2. Is the manuscript technically sound, and do the data support the conclusions?

Reviewer #1: Partly

3. Has the statistical analysis been performed appropriately and rigorously? 

Reviewer #1: Yes

4. Have the authors made all data underlying the findings in their manuscript fully available?

Reviewer #1: No

5. Is the manuscript presented in an intelligible fashion and written in standard English?

Reviewer #1: No

6. Review Comments to the Author

Reviewer #1: Comments: Though the manuscript has been improved significantly but English is still of sub-standard, which needs to be improved. I have pin-pointed few instances, which are as follows:

1. The word ‘rates of disease’ in Abstract and otherwise need to be changed, e.g. ‘outcome of disease’. At few instances, rate of disease is OK but at few instances, it reads really odd.

2. Tuberculosis need to be abbreviated as TB first time and later, TB can be written in Abstract as these two terms are written haphazardly.

3. Lines, 261-68 of Discussion need to be rewritten. Similarly, line 269 needs attention.

4. ‘incidence rates (lines 273-274 of Discussion) for extrapulmonary TB and TB-HIV co-infection changed in structure after 2014’,the word ‘structure’ needs to be replaced. Similarly, sentence in lines 275-276 should be rewritten. It should be ‘might’ instead of ‘may’ in line 278.

5. Line 280, in Discussion, appropriate reference shall be given. While interpreting pulmonary and extrapulmonary TB samples by molecular test (GeneXpert), sensitivity and specificity may be almost similar in the given reference, but mostly pulmonary TB exhibited a much higher sensitivity than exptrapulmonary TB. Pl quote those references and give valid reasons for that, e.g. due to paucibacillary nature of specimens, etc.

6. Lines, 284-90, shall be rewritten. What do you mean positive in people who actually have TB? I must point out that GeneXpert mostly does not perform well with Pleural TB cases as it reveals poor sensitivity (Pl see Review of i) Denkinger CM, Schumacher SG, Boehme CC, Dendukuri N, Pai M, Steingart KR. Xpert MTB/RIF assay for the diagnosis of extrapulmonary tuberculosis: a systematic review and meta-analysis. Eur. Respir. J. 44, 435–446 (2014). ii) Sehgal IS, Dhooria S, Aggarwal AN, Behera D, Agarwal R. Diagnostic performance of Xpert MTB/RIF in tuberculous pleural effusion: systematic review and meta-analysis. J. Clin. Microbiol. 54(4), 1133–1136 (2016).

7. Lines 297-301, also, write that culture is labor-intensive method and requires skillful technicians. You are probably mentioning culture on LJ medium. What about culture on MGIT-960 system? Write plus and negative points for that.

8. Line 306 of Discussion, what do you mean by ‘consequent change in structure in the time series’? Please write clearly. Similarly, lines 322-323 need attention.

9. Line 326, further studies in which direction will improve the understanding of this test? There is probably no need to further understand the test. Yes, we can probably improve the sensitivity by using Xpert Ultra. Also, may be the technology could be improved as in case of Xpert Omni. You need to mention these points.

10. Line 328, ‘studies of this nature’ need to be replaced. Similarly, lines 335-338 need to be rewritten.

7. PLOS authors have the option to publish the peer review history of their article (what does this mean?). If published, this will include your full peer review and any attached files.

Reviewer #1: **Yes: **Promod Mehta

---

## [Author Response · Author response to Decision Letter 1]

10 Mar 2021

Dear Editor, 

Many thanks for your reply and your reviewers' comments about our manuscript “PONE-D-20-34732 Effectiveness and trend forecasting of tuberculosis diagnosis after the introduction of GeneXpert® in a city in south-eastern Brazil”.

The comments were appropriate to qualify/improve the manuscript. We have forwarded a letter with the changes made in the manuscript, presenting our response for each comment from reviewers, as requested. Many thanks for the comments.

Kind regards,

Berra et al.

Reviewers' comments:

Reviewer #1: 

1. The word ‘rates of disease’ in Abstract and otherwise need to be changed, e.g. ‘outcome of disease’. At few instances, rate of disease is OK but at few instances, it reads really odd.

Response: Thank you for your comment. In this case of the objective of the study, what we did from the temporal modeling was to make a prediction of the incidence rates of tuberculosis, therefore, we believe that substituting for "outcome of disease" will not keep the same meaning of the phrase, since in the present study we do not assess the outcome of tuberculosis, but work with the rates.

2. Tuberculosis need to be abbreviated as TB first time and later, TB can be written in Abstract as these two terms are written haphazardly.

Response: Thank you for your comment. As suggested, we introduced the word tuberculosis in full the first time it appears and replaced it with its acronym (TB) throughout the Abstract.

3. Lines, 261-68 of Discussion need to be rewritten. Similarly, line 269 needs attention.

Response: Thank you for your comment. Paragraphs have been modified to better understand their content.

4. ‘incidence rates (lines 273-274 of Discussion) for extrapulmonary TB and TB-HIV co-infection changed in structure after 2014’,the word ‘structure’ needs to be replaced. Similarly, sentence in lines 275-276 should be rewritten. It should be ‘might’ instead of ‘may’ in line 278.

Response: Thank you for your comment. The paragraph was rewritten but we chose not to remove the word "structure" since it is related to the method used and, in this way, we consider that it facilitates the reader to relate the result presented with the method used.

5. Line 280, in Discussion, appropriate reference shall be given. While interpreting pulmonary and extrapulmonary TB samples by molecular test (GeneXpert), sensitivity and specificity may be almost similar in the given reference, but mostly pulmonary TB exhibited a much higher sensitivity than exptrapulmonary TB. Pl quote those references and give valid reasons for that, e.g. due to paucibacillary nature of specimens, etc.

Response: Thank you for your comment. I apologize but I am not sure if I understood correctly what was suggested, but we have added two paragraphs to the discussion on this issue of extrapulmonary samples.

6. Lines, 284-90, shall be rewritten. What do you mean positive in people who actually have TB? I must point out that GeneXpert mostly does not perform well with Pleural TB cases as it reveals poor sensitivity (Pl see Review of i) Denkinger CM, Schumacher SG, Boehme CC, Dendukuri N, Pai M, Steingart KR. Xpert MTB/RIF assay for the diagnosis of extrapulmonary tuberculosis: a systematic review and meta-analysis. Eur. Respir. J. 44, 435–446 (2014). ii) Sehgal IS, Dhooria S, Aggarwal AN, Behera D, Agarwal R. Diagnostic performance of Xpert MTB/RIF in tuberculous pleural effusion: systematic review and meta-analysis. J. Clin. Microbiol. 54(4), 1133–1136 (2016).

Response: Thank you for your comment. We rewrote the sentence in order to make it clear that we are talking about false positive and negative results and added a note about the lower effectiveness of the test for pleural samples. Thanks for sending the references, they were added and cited in the text.

7. Lines 297-301, also, write that culture is labor-intensive method and requires skillful technicians. You are probably mentioning culture on LJ medium. What about culture on MGIT-960 system? Write plus and negative points for that.

Response: Thank you for your comment. We inserted a paragraph in the discussion comparing the results of the culture carried out in solid medium and automated liquid medium.

8. Line 306 of Discussion, what do you mean by ‘consequent change in structure in the time series’? Please write clearly. Similarly, lines 322-323 need attention.

Response: Thanks for your comment, the phrases have been revised and rewritten to make your content clearer.

9. Line 326, further studies in which direction will improve the understanding of this test? There is probably no need to further understand the test. Yes, we can probably improve the sensitivity by using Xpert Ultra. Also, may be the technology could be improved as in case of Xpert Omni. You need to mention these points.

Response: Thank you for your comment. We added a paragraph about GeneXpert Ultra and Omni.

10. Line 328, ‘studies of this nature’ need to be replaced. Similarly, lines 335-338 need to be rewritten. 

Response: Thank you for your comment. We changed the word "nature" to "design" and the last paragraph of the conclusion has been rewritten, as suggested.

---

## [Decision Letter · Decision Letter 2]

26 Mar 2021

PONE-D-20-34732R2

Effectiveness and trend forecasting of tuberculosis diagnosis after the introduction of GeneXpert® in a city in south-eastern Brazil

PLOS ONE

Dear Dr. Berra,

Thank you for submitting your manuscript to PLOS ONE. After careful consideration, we feel that it has merit but does not fully meet PLOS ONE’s publication criteria as it currently stands. Therefore, we invite you to submit a revised version of the manuscript that addresses the points raised during the review process.

Please submit your revised manuscript. If you will need significantly more time than this to complete your revisions, please reply to this message or contact the journal office at plosone@plos.org. Please include the following items when submitting your revised manuscript:

We look forward to receiving your revised manuscript.

Kind regards,

Frederick Quinn

Academic Editor

PLOS ONE

Journal Requirements:

Reviewers' comments:

Reviewer's Responses to Questions

**Comments to the Author**

1. If the authors have adequately addressed your comments raised in a previous round of review and you feel that this manuscript is now acceptable for publication, you may indicate that here to bypass the “Comments to the Author” section, enter your conflict of interest statement in the “Confidential to Editor” section, and submit your "Accept" recommendation.

Reviewer #1: All comments have been addressed

2. Is the manuscript technically sound, and do the data support the conclusions?

Reviewer #1: Partly

3. Has the statistical analysis been performed appropriately and rigorously? 

Reviewer #1: Yes

4. Have the authors made all data underlying the findings in their manuscript fully available?

Reviewer #1: No

5. Is the manuscript presented in an intelligible fashion and written in standard English?

Reviewer #1: No

6. Review Comments to the Author

Reviewer #1: Manuscript is improved but still there are pitfalls. English needs to be improved tremendously so that it reads well.

1. Conclusions section of Abstract need to be rewritten.

2. HIV on lines 51 and 53. First time you wrote HIV, second time you write human immunodeficiency virus (HIV). Not correct. In fact, writing HIV itself is fine at both the places as this term so common. Similarly, when you write first time, Mycobacterium tuberculosis is fine, but later when you write M. tuberculosis is fine.

3. Line 60, it should be ‘prolonged turnaround time’.

4. Line 74, it is highly sensitive for pulmonary TB, but not for extrapulmonary TB.

5. Lines 282-86, need to write clearly that when culture was employed as the gold standard, then the sensitivity and specificity for GeneXpert were determined for pulmonary and extrapulmonary TB. Please read this Ref 23 carefully and then write.

6. Lines 288-89, need to specifically mention here GeneXpert assay and not molecular tests because there are many other molecular tests. Moreover, at times with many extrapulmonary TB samples, GeneXpert shows poor sensitivity so ‘good test performance’ shall be replaced.

7. Lines 294-95 shall be rewritten.

8. Line 300, it shall be written as ‘not good sensitivity with Xpert assay’ instead of efficacy with molecular test. This is because there are many other molecular tests.

9. Line 311 ‘performing the exam’ is not appropriate. Pl re-write. In fact lines 311-13 need to be rewritten.

10. Line please elaborate about liquid culture e.g. MGIT-960 and its advantages over sold media.

11. Lines 330-331 still not clear, please rewrite.

12. Line 374, what are those certain ways?

13. Lines 374-76, need to be rewritten.

7. PLOS authors have the option to publish the peer review history of their article (what does this mean?). If published, this will include your full peer review and any attached files.

Reviewer #1: No

---

## [Author Response · Author response to Decision Letter 2]

5 Apr 2021

Dear Editor, 

Many thanks for your reply and your reviewers' comments about our manuscript “PONE-D-20-34732 Effectiveness and trend forecasting of tuberculosis diagnosis after the introduction of GeneXpert® in a city in south-eastern Brazil”.

The comments were appropriate to qualify/improve the manuscript. We have forwarded a letter with the changes made in the manuscript, presenting our response for each comment from reviewers, as requested. Many thanks for the comments.

Kind regards,

Berra et al.

Editors’ comments:

Response: Thank you for your comment. All references have been revised.

Reviewers' comments:

Reviewer #1: 

Manuscript is improved but still there are pitfalls. English needs to be improved tremendously so that it reads well.

Response: Thank you for reevaluating the manuscript once again. We inform that the manuscript had been revised in English by a specialized company (https://www.proof-reading-service.com) by natives. Again we send the manuscript to have its writing corrected, we hope it has improved.

1. Conclusions section of Abstract need to be rewritten.

Response: The conclusion of Abstract has been modified.

2. HIV on lines 51 and 53. First time you wrote HIV, second time you write human immunodeficiency virus (HIV). Not correct. In fact, writing HIV itself is fine at both the places as this term so common. Similarly, when you write first time, Mycobacterium tuberculosis is fine, but later when you write M. tuberculosis is fine.

Response: In fact, the first time that the acronym HIV appears is in the first paragraph of the introduction, along with the acronym AIDS, in which both acronyms were written in full.

In the lines in which you cited, the second appearance was spelled out in full because it is a new acronym, TB-HIV co-infection, therefore, just being the union of two acronyms that have already appeared in the text, we write its meaning in full in order to follow the journal's norms.

Regarding Mycobacterium tuberculosis, we leave it in full the first time it appears in the text and at other times we change it to M. tuberculosis, as recommended.

3. Line 60, it should be ‘prolonged turnaround time’.

Response: Thanks, this has been fixed.

4. Line 74, it is highly sensitive for pulmonary TB, but not for extrapulmonary TB.

Response: Thanks, this has been fixed.

5. Lines 282-86, need to write clearly that when culture was employed as the gold standard, then the sensitivity and specificity for GeneXpert were determined for pulmonary and extrapulmonary TB. Please read this Ref 23 carefully and then write.

Response: Thank you. that information was inserted in the same paragraph.

6. Lines 288-89, need to specifically mention here GeneXpert assay and not molecular tests because there are many other molecular tests. Moreover, at times with many extrapulmonary TB samples, GeneXpert shows poor sensitivity so ‘good test performance’ shall be replaced.

Response: Thank you, the paragraph has been rewritten.

7. Lines 294-95 shall be rewritten.

Response: Thank you, the paragraph has been rewritten.

8. Line 300, it shall be written as ‘not good sensitivity with Xpert assay’ instead of efficacy with molecular test. This is because there are many other molecular tests.

Response: Thanks, this has been fixed.

9. Line 311 ‘performing the exam’ is not appropriate. Pl re-write. In fact lines 311-13 need to be rewritten.

Response: Thank you, the paragraph has been rewritten.

10. Line please elaborate about liquid culture e.g. MGIT-960 and its advantages over sold media.

Response: Three paragraphs were added to the discussion session, better addressing the culture in liquid medium and also comparing it with solid medium L-J.

11. Lines 330-331 still not clear, please rewrite.

Response: The sentence was rewritten again.

12. Line 374, what are those certain ways?

Response: Thanks, this has been fixed.

13. Lines 374-76, need to be rewritten.

Response: Thank you, the paragraph has been rewritten.

---

## [Decision Letter · Decision Letter 3]

23 Apr 2021

PONE-D-20-34732R3

Effectiveness and trend forecasting of tuberculosis diagnosis after the introduction of GeneXpert® in a city in south-eastern Brazil

PLOS ONE

Dear Dr. Berra,

Thank you for submitting your manuscript to PLOS ONE. After careful consideration, we feel that it has merit but does not fully meet PLOS ONE’s publication criteria as it currently stands. Therefore, we invite you to submit a revised version of the manuscript that addresses the points raised during the review process.

Please submit your revised manuscript. If you will need significantly more time to complete your revisions, please reply to this message or contact the journal office at plosone@plos.org. Please include the following items when submitting your revised manuscript:

We look forward to receiving your revised manuscript.

Kind regards,

Frederick Quinn

Academic Editor

PLOS ONE

Journal Requirements:

Reviewers' comments:

Reviewer's Responses to Questions

**Comments to the Author**

1. If the authors have adequately addressed your comments raised in a previous round of review and you feel that this manuscript is now acceptable for publication, you may indicate that here to bypass the “Comments to the Author” section, enter your conflict of interest statement in the “Confidential to Editor” section, and submit your "Accept" recommendation.

Reviewer #1: All comments have been addressed

2. Is the manuscript technically sound, and do the data support the conclusions?

Reviewer #1: Partly

3. Has the statistical analysis been performed appropriately and rigorously? 

Reviewer #1: Yes

4. Have the authors made all data underlying the findings in their manuscript fully available?

Reviewer #1: Yes

5. Is the manuscript presented in an intelligible fashion and written in standard English?

Reviewer #1: No

6. Review Comments to the Author

Reviewer #1: Comments

1. Conclusion of Abstract, line 45, it shall be ‘it is possible’, instead of ‘is it possible’.

2. In Intro, line 53, overcoming AIDS today, you need to reframe this sentence.

3. Line 63, shall be ‘considered as the gold standard’.

4. Line 72, it shall be based on the real-time PCR, instead of ‘polymerase chain reaction (PCR), also known as real-time PCR’. This is because it is incorrect to write that PCR is also known as real-time PCR, two techniques are different.

5. Line 80, GeneXpert, in general, is not considered to be a cost-effective test especially in developing countries. Rather, it is considered as a major drawback of this technique as it is not cost-effective.

6. Lines, 77 and 84, as well as 259, it shall be ‘for the detection of’ (and not ‘in the detection of ’).

7. Line 270, it shall be ‘decreases to 57 - 78.2%’ instead of drops to 57 to 78.2%.

8. Lines, 270-274 shall be split into two sentences and written with clarity.

9. Lines 296 and 298 as well as line 300, it shall be sputum samples and not lung samples.

10. Lines 301-303 are incorrect. ‘it is important that further studies are carried out with this type of material as this result may change according to the number of samples tested’, need to be re-written. It is not only the sample size, but different nature of extrapulmonary TB specimens and their collection across diverse geographical locations of entire world.

11. Lines 305 and 311 are repeated (‘Thus, the performance of the GeneXpert® system in extrapulmonary samples is still not entirely clear’).

12. Line 311, ‘again we should stress’, it shall be rewritten.

13. Line 316, ‘to confirm your diagnosis’, is wrong. Please omit the word ‘your’ here.

14. Lines 327, sputum smear and culture examinations are for pulmonary TB, only. You need to mention. That. For extrapulmonary TB, sputum is not taken; rather other bodily fluids undergo smear / culture examination.

15. Lines 354 and 355, two times high cost. Please rewrite.

7. PLOS authors have the option to publish the peer review history of their article (what does this mean?). If published, this will include your full peer review and any attached files.

Reviewer #1: No

---

## [Author Response · Author response to Decision Letter 3]

29 Apr 2021

Dear Editor, 

Many thanks for your reply and your reviewers' comments about our manuscript “PONE-D-20-34732 Effectiveness and trend forecasting of tuberculosis diagnosis after the introduction of GeneXpert® in a city in south-eastern Brazil”.

The comments were appropriate to qualify/improve the manuscript. We have forwarded a letter with the changes made in the manuscript, presenting our response for each comment from reviewers, as requested. Many thanks for the comments.

Kind regards,

Berra et al.

Editors’ comments:

Response: Thank you for your comment. All references have been revised.

Reviewers' comments:

Reviewer #1: 

1. Conclusion of Abstract, line 45, it shall be ‘it is possible’, instead of ‘is it possible’.

Response: Thank you, this has been fixed.

2. In Intro, line 53, overcoming AIDS today, you need to reframe this sentence.

Response: Thanks for your comment, this sentence has been modified to improve your understanding.

3. Line 63, shall be ‘considered as the gold standard’.

Response: Thank you, this has been fixed.

4. Line 72, it shall be based on the real-time PCR, instead of ‘polymerase chain reaction (PCR), also known as real-time PCR’. This is because it is incorrect to write that PCR is also known as real-time PCR, two techniques are different.

Response: Thank you, this has been fixed.

5. Line 80, GeneXpert, in general, is not considered to be a cost-effective test especially in developing countries. Rather, it is considered as a major drawback of this technique as it is not cost-effective.

Response: Thank you for your comment, we put this observation at the end of that same indicated paragraph.

6. Lines, 77 and 84, as well as 259, it shall be ‘for the detection of’ (and not ‘in the detection of ’).

Response: Thank you, this has been fixed.

7. Line 270, it shall be ‘decreases to 57 - 78.2%’ instead of drops to 57 to 78.2%.

Response: Thank you, this has been fixed.

8. Lines, 270-274 shall be split into two sentences and written with clarity.

Response: Thanks for your comment, the paragraph has been rewritten.

9. Lines 296 and 298 as well as line 300, it shall be sputum samples and not lung samples.

Response: Thank you, this has been fixed.

10. Lines 301-303 are incorrect. ‘it is important that further studies are carried out with this type of material as this result may change according to the number of samples tested’, need to be re-written. It is not only the sample size, but different nature of extrapulmonary TB specimens and their collection across diverse geographical locations of entire world.

Response: Thank you for your comment. These observations have been inserted in the paragraph.

11. Lines 305 and 311 are repeated (‘Thus, the performance of the GeneXpert® system in extrapulmonary samples is still not entirely clear’).

Response: Thank you, this has been fixed.

12. Line 311, ‘again we should stress’, it shall be rewritten.

Response: Thank you, this has been fixed.

13. Line 316, ‘to confirm your diagnosis’, is wrong. Please omit the word ‘your’ here.

Response: Thank you, this has been fixed.

14. Lines 327, sputum smear and culture examinations are for pulmonary TB, only. You need to mention. That. For extrapulmonary TB, sputum is not taken; rather other bodily fluids undergo smear / culture examination.

Response: Thank you for your comment. This observation was inserted in the paragraph in question.

15. Lines 354 and 355, two times high cost. Please rewrite.

Response: Thank you, this has been fixed.

---

## [Decision Letter · Decision Letter 4]

17 May 2021

Effectiveness and trend forecasting of tuberculosis diagnosis after the introduction of GeneXpert® in a city in south-eastern Brazil

PONE-D-20-34732R4

Dear Dr. Berra,

We’re pleased to inform you that your manuscript has been judged scientifically suitable for publication and will be formally accepted for publication once it meets all outstanding technical requirements.

Kind regards,

Frederick Quinn

Academic Editor

PLOS ONE

Additional Editor Comments (optional):

Reviewers' comments:

Reviewer's Responses to Questions

**Comments to the Author**

1. If the authors have adequately addressed your comments raised in a previous round of review and you feel that this manuscript is now acceptable for publication, you may indicate that here to bypass the “Comments to the Author” section, enter your conflict of interest statement in the “Confidential to Editor” section, and submit your "Accept" recommendation.

Reviewer #1: All comments have been addressed

2. Is the manuscript technically sound, and do the data support the conclusions?

Reviewer #1: Partly

3. Has the statistical analysis been performed appropriately and rigorously? 

Reviewer #1: Yes

4. Have the authors made all data underlying the findings in their manuscript fully available?

Reviewer #1: Yes

5. Is the manuscript presented in an intelligible fashion and written in standard English?

Reviewer #1: No

6. Review Comments to the Author

Reviewer #1: Comments

1. Lines 299-302 and lines 303-312 are repetition; therefore lines 299-302 shall be omitted in discussion.

Moreover, lines 303-312 need to be rewritten, as English is of substandard.

2. Line 351: the term MGIT-960 System shall be written above along with automated liquid culture system in lines 344-345.

7. PLOS authors have the option to publish the peer review history of their article (what does this mean?). If published, this will include your full peer review and any attached files.

Reviewer #1: No

---

## [Editor Report · Acceptance letter]

19 May 2021

PONE-D-20-34732R4 

Effectiveness and trend forecasting of tuberculosis diagnosis after the introduction of GeneXpert in a city in south-eastern Brazil 

Dear Dr. Berra:

I'm pleased to inform you that your manuscript has been deemed suitable for publication in PLOS ONE. Congratulations! Your manuscript is now with our production department. 

Kind regards, 

on behalf of

Dr. Frederick Quinn 

Academic Editor

PLOS ONE